# Agricultural and Biomedical Applications of Chitosan-Based Nanomaterials

**DOI:** 10.3390/nano10101903

**Published:** 2020-09-24

**Authors:** Subhani Bandara, Hongbo Du, Laura Carson, Debra Bradford, Raghava Kommalapati

**Affiliations:** 1Cooperative Agricultural Research Center, Prairie View A&M University, Prairie View, TX 77446, USA; lecarson@pvamu.edu (L.C.); dabradford@pvamu.edu (D.B.); 2Center for Energy and Environmental Sustainability, Prairie View A&M University, Prairie View, TX 77446, USA; hodu@pvamu.edu (H.D.); rrkommalapati@pvamu.edu (R.K.); 3Department of Civil and Environmental Engineering, Prairie View A&M University, Prairie View, TX 77446, USA

**Keywords:** chitosan, nanoparticles, abiotic stress, water purification, foodborne pathogens, cancer photothermal therapy

## Abstract

Chitosan has emerged as a biodegradable, nontoxic polymer with multiple beneficial applications in the agricultural and biomedical sectors. As nanotechnology has evolved as a promising field, researchers have incorporated chitosan-based nanomaterials in a variety of products to enhance their efficacy and biocompatibility. Moreover, due to its inherent antimicrobial and chelating properties, and the availability of modifiable functional groups, chitosan nanoparticles were also directly used in a variety of applications. In this review, the use of chitosan-based nanomaterials in agricultural and biomedical fields related to the management of abiotic stress in plants, water availability for crops, controlling foodborne pathogens, and cancer photothermal therapy is discussed, with some insights into the possible mechanisms of action. Additionally, the toxicity arising from the accumulation of these nanomaterials in biological systems and future research avenues that had gained limited attention from the scientific community are discussed here. Overall, chitosan-based nanomaterials show promising characteristics for sustainable agricultural practices and effective healthcare in an eco-friendly manner.

## 1. Introduction

In the realm of climate change, increasing population, and the decrease in the land that can be cultivated, agriculture and health systems are facing numerous challenges. Nanotechnology can play an essential role in addressing these issues by promoting enhanced crop production, optimum usage of the land, and the creation of advanced drugs. The small size of the nanomaterials is advantageous in crossing the biological barriers and carrying the required molecules into various locations in animals and plants. However, the nanoparticle size should be optimized based on the intended application to minimize toxic side effects, as discussed under Section 4 of this review. When properly used, nanoparticles engineered from chitosan and its derivatives can be indispensable in addressing issues related to feeding the increasing population and improving healthcare.

Chitosan is a linear copolymer with *D*–glucosamine and *N*–acetyl-*D*–glucosamine units joined via β–(1–4) glycosidic bonds (Figure 1) and has been extensively used for the production of nanoparticles by researchers, as reviewed in this article. Chitosan has been popular due to its antimicrobial, antioxidant, and chelating properties, together with its nontoxic and biocompatible nature [1]. As a cationic polymer, chitosan inherently possesses bio-adhesion, cellular transfection, anti-inflammatory, and anti-hypercholesterolemic characteristics, which can be enhanced by combining with other materials, making it an attractive candidate for biomedical and agricultural applications [2]. Chitosan is also an excellent carrier for nanoparticles due to its ability to penetrate across cellular barriers and to flow through narrow intercellular junctions in epithelial cells [3]. The availability of hydroxyl and amino groups on chitosan provides an excellent platform for complexation with other molecules/compounds and helps to transform them into more stable complexes with better pharmacokinetic properties [3]. Additionally, due to the availability of functional groups, chitosan can be modified in different ways to obtain substituted, crosslinked, carboxylated, ionic and bounded derivatives to match the various research needs [2]. The different methods utilized in synthesizing chitosan-based nanoparticles, such as emulsion crosslinking, emulsion-droplet coalescence, ionotropic gelation, reverse micellisation, and precipitation, have been described in detail in the literature [4,5].

Chitosan is obtained from a variety of sources by deacetylation of chitin and contains more than 7% nitrogen and less than 40% degree of acetylation [6]. Although the primary commercial production source for chitosan is crustacean shells, their seasonal availability and the use of harsh chemicals in the production, and the generation of large amounts of alkaline waste had led to environmental issues as well as product quality variability [6]. Furthermore, the use of crustacean sources can limit the application of the polymer, specifically in the biomedical field, due to the associated shellfish allergies. Hence, researchers have looked into different sources for the production of chitosan. Shanmuganathan et al. have extensively reviewed a variety of sources for the extraction of chitosan, including various species of microorganisms, insects, and other aquatic animals [7]. Additionally, chitinous cell wall material was isolated from yeast species *Rhodosporidium paludigenum* and *Saccharomyces cerevisiae* [8]. Fungal mycelium of *Allomyces arbuscula*, *Mucor genevensis*, *Tranetes versicolor,* and the fruiting body of *Agaricus bisporus* were also utilized as more cost-effective and renewable sources for chitin compared to crustacean shells [9]. A pest attacking agricultural crop in Mexico and Central America, *Schistocerca piceifrons piceifrons* (Orthoptera: Acrididae) has also been used for the successful extraction of chitin and chitosan at 11.88% and 9.11% yield, respectively [10]. These sources undergo demineralization, deproteinization, and decolorization to produce chitin, and it is converted to chitosan via different chemical and enzymatic deacetylation methods as described by Shanmuganathan et al. (Figure 1). 

## 2. Applications in Agriculture

Chitosan-based nanoparticles (CNPs) have been used in agriculture as pesticides, herbicides, insecticides, and to obtain better quality food products with a higher yield, and many of these applications of chitosan have been extensively reviewed in the literature [11]. Nano-chitosan based materials or chitosan combined with other nanoparticles were applied to preserve fresh fruits such as strawberries [12], Jujube [13], loquat [14], and longan [15] during storage. Additionally, chitosan can serve as an encapsulating agent by itself and in combination with other materials in the production of slow-release fertilizers, owing to its cationic nature, biodegradability, non-toxicity, and adsorption properties [4,16]. This section of the review is focused on areas that have gained limited attention of the scientific community related to the usage of chitosan-based nanomaterials, which include water purification for agricultural uses and managing abiotic stress in plants.

### 2.1. Water Purification and Sustainable Agriculture

Due to drought and rapid development of industry in some regions, people have to look for alternative water resources, such as seawater and non-conventional water, for agricultural irrigation. Non-conventional water can be wastewater emitted from domestic, municipal, and industrial sewer, and saline water from salt lakes and shale oil and gas industry. To maintain crop health and to avoid soil damage from the direct use of wastewater in agriculture, it is essential to treat the wastewater sufficiently to meet the requirements of crop growth. To seek sustainable approaches for wastewater treatment and water purification, products derived from renewable materials, such as activated carbon, cellulose, and chitosan, are good options at low cost and high energy efficiency [17,18,19]. 

Microstructures, nanoparticles, and nanocomposites of chitosan have widely been used as an absorbent to remove various inorganic and organic pollutants as rich hydroxyl and amino groups are present in the crosslinked structure of chitosan [20,21]. Compared to its microparticles, the larger surface areas of chitosan-based nanoparticles introduce more efficient adsorption capacity for dyes, pesticides, phosphate, and heavy metal ions such as Pb(II), Hg(II), Cd(II), Cr(III), Cr(VI), Cu(II), Co(II), Ni(II) and rare earth metals [22,23,24]. The recycling of adsorbent is of great importance economically in water treatment, and magnetic chitosan nanoparticles made from Fe_3_O_4_ are very promising to reduce the operation cost by recycling the adsorbent under a magnetic field. Chitosan-based nanoparticles can also be prepared as bioflocculants to settle down pollutant particles during the flocculation step in a series of water treatment processes [25,26]. An example of chitosan-grafted magnetic nanoparticles used for oily wastewater treatment is demonstrated in Figure 2. In some advanced wastewater treatments using membranes, some chitosan nanoparticles are embodied into ultrafiltration and nanofiltration membranes to reduce chemical oxygen demands, color, metal ions, and to enhance the antifouling properties [27,28,29]. Various applications of chitosan nanoparticles in drinking water purification and agricultural/industrial wastewater treatment are summarized in Table 1. 

### 2.2. Applications of Nanochitosan in Regulating Abiotic Stress in Plants

#### 2.2.1. Nanochitosan in Controlling Salinity Stress

Salinity is a significant stress factor affecting plant growth throughout the world. From all the agricultural lands in the world, more than 20% is stipulated to have high salinity at a level that can negatively affect plant growth [49]. Additionally, by 2050, around 50% of agricultural lands are predicted to be affected by salinity [50]. Bean plant (*Phaseolus vulgaris* L.) belonging to the salt-sensitive classification has shown improved seed germination when treated with 0.1%, 0.2%, and 0.3% nanochitosan at 100 Mm salt concentration [49]. Nitric oxide (NO) releasing chitosan nanoparticles were shown to be more effective than free NO donors in combating salt stress in maize [51] (Figure 3). Chitosan delivered NO has enhanced the leaf S-nitrosothiols, photosystem II activity, and chlorophyll levels in treated plants by increasing the bioavailability of NO [51]. Solid matrix priming of mung bean seedlings with nanochitosan had shown to reduce the harmful effects of salinity stress and improve metabolism, growth, protein levels, and chlorophyll content of the plants [50]. Chitosan-polyvinyl alcohol hydrogels with and without copper nanoparticles applied to tomato plants under salt stress were shown to promote the plant growth and elevate the expression of genes for the production of jasmonic acid (JA) and superoxide dismutase (SOD), which are necessary for detoxification [52]. During previous research, chitosan in bulk form was shown to alleviate the effects of salt stress for wheat [53], chickpea [54], lentils [55], and ajowan seeds [56]. Nanochitosan could possibly have a more significant effect on these crops due to its high surface to volume ratio resulting in higher penetrability and the ability to form more interactions.

#### 2.2.2. Nanochitosan in Controlling Drought Stress

Water deficiency can affect many aspects of a plant, including anatomy, physiology, and biochemistry leading to lower yields [57]. Drought stress is known to destroy the chloroplasts, reducing chlorophyll content and the activity of the enzymes involved in the Calvin cycle of photosynthesis [58]. Additionally, drought stress causes the closure of stomata, blocking the CO_2_ intake by the leaves, further leading to the reduction of photosynthesis and plant growth [58]. In a study conducted by Behboudi et al. (2018) with barley plants, application of chitosan nanoparticles at 60 and 90 ppm concentrations via the soil or foliar routes, were shown to eliminate harmful effects of late-season drought stress related to relative water content (RWC), plant growth and yield [59]. Foliar application of nanochitosan emulsion on pearl millet under drought stress was shown to enhance the water status of plants by decreasing stomatal conductance and transpiration [60]. Nanochitosan encapsulating S-nitrosoglutathione (NO donor) was shown to alleviate the effects of drought stress in sugarcane plants [61]. The sugarcane plants exhibited a higher photosynthesis rate and root biomass compared to the ones treated with free S-nitrosoglutathione, showing the controlled release of NO by nanochitosan is more effective in combating drought [61]. The application of chitosan nanoparticles at 90 ppm concentration on water-deficient wheat plants through soil or leaves had improved both the biochemical and physiological characteristics of the plants [62]. Additionally, bulk chitosan was shown to promote the growth of apple explants under drought stress induced by agar, at 40 mg/L concentration [63]. Additionally, foliar application of bulk chitosan to wheat plants under water stress has lowered the deleterious effects on plant growth and yield [57]. Rabelo et al. (2019) reported that the foliar application of N-succinyl chitosan and N,O-dicarboxymethylated chitosan derivatives had increased the tolerance of water stress in a hybrid maize species that is sensitive to drought stress [64]. These chitosan derivatives were shown to induce the antioxidant defense system, increase the production of phenolic compounds, osmoregulators, and crop yield, and promote gas exchange in leaves [64]. Hence, it is possible that nanochitosan derivatives also will have a positive effect on the growth of these plants under drought stress.

#### 2.2.3. The Potential Use of Nanochitosan under Temperature and Heavy Metal Stress

Low and high-temperature stress and metal contaminated soil are two other common factors affecting Agricultural land use. Although recent research on the use of nanochitosan to alleviate the stress caused by heat and heavy metals are scarce, there are reports on the use of bulk chitosan. Ibrahim and Ramadan had reported the productive use of bulk chitosan and zinc to reduce the heat stress on late sowing dry bean (*Phaseolus vulgaris* L.) [65]. Additionally, priming the maize seeds with chitosan at low temperature (15 °C) has shown to increase the germination index, shoot height, root length, and dry root weight and decrease the time taken for germination [66]. Furthermore, a study conducted by Kamari et al. (2011) revealed the ability of chitosan and its derivatives to effectively complex metal ions (Ag(I), Pb(II), and Cu(II)) in soil that co-exist with other ionic substances like K^+^, Cl^−^ and NO_3_^−^ [67]. Hence, the use of nanochitosan to overcome both high and low-temperature stress and to relieve metal toxicity in the soil are fields that can be evaluated in future research.

#### 2.2.4. Mechanism of Action of Chitosan Nanoparticles in Combating Abiotic Stresses

The abiotic stress response of plants caused by low and high temperatures, high salinity, drought, or other conditions may have some shared attributes, although the primary stimuli are different. During signal transduction, the primary signal activates the production of secondary signaling molecules such as inositol phosphate and reactive oxygen species (ROS) resulting in the receptor-mediated release of Ca^2+^ in the cells. The events mentioned above can cause phosphorylation driven modulation of specific proteins that have a direct protective function or act as transcription factors to regulate the expression of the genes involved in stress response [68]. According to Xiong et al., the initial stress signal may be perceived by several primary sensors leading to secondary signals and downstream signaling cascades that may occur at a different time and location from the primary signaling [68]. Therefore, these secondary signals may be shared between different stress response pathways providing “stress cross-protection” to the plants [68]. Hence, the involvement of a single compound such as chitosan to alleviate the adverse effects of different abiotic stresses may occur at the points where these different pathways interact with each other. High heat, drought, and high salinity in soil were all known to affect metabolic processes such as photosynthesis and protein synthesis, resulting in decreased growth rate and lower quality of crops [49,69]. High heat can negatively affect photosynthetic carbon fixation and redistribution, as well as the electron transport chain occurring in the chloroplasts [69]. Zhang et al. reported that during drought stress, plants act by increased stomatal closure and reduced activity of photosynthetic enzymes such as ribulose-1,5-bisphosphate carboxylase/oxygenase (Rubisco) [69]. Additionally, as a recovery method to combat drought stress, plants increase the production of solutes such as amino acids, polyols, and sugars to mediate the turgor pressor. Barley plants treated with nano-chitosan under drought have shown elevated levels of the osmoprotective amino acid proline [59]. Salt stress is also known to increase the accumulation of numerous compounds like proline [53]. Moreover, nanochitosan application on maize has elevated the levels of organic compounds such as phenols, aldehydes, ketones, etc. that are acting as stress tolerance regulators [70]. Additionally, plants synthesize more antioxidants and antioxidative enzymes to neutralize the higher levels of ROS produced during stress as a result of the disrupted electron transport chain and express specific proteins like late embryogenesis abundant protein that are known to have a defensive function [59,71]. Nanochitosan was shown to enhance the activity of antioxidative catalase (CAT) and superoxide dismutase (SOD) enzymes during drought stress in Barley [59]. SOD functions to form H_2_O_2_ by neutralizing superoxide free radicals, and it is broken down into nontoxic water and oxygen by the action of CAT. Hence nanochitosan may act by inducing the antioxidant enzymes in plants and by regulating the levels of osmoprotectants like proline [59]. Additionally, the chitosan-based increase in proline levels was reported to be linked to the enhanced activity of proteinase enzymes [58]. Additionally, bulk chitosan was shown to display abscisic acid (ABA) dependent antitranspirant activity due to the induction of stomatal closure [71]. ABA is a well-known plant signaling molecule that accumulates in response to various stress conditions, regulating stomatal closure and expression of stress-responsive genes [69]. Hence, it is possible that nanochitosan is also involved in an ABA-dependent mechanism in controlling abiotic stress in plants.

## 3. Biomedical Applications of Nanochitosan

Chitosan is preferably used in the biomedical field due to its favorable properties such as biocompatibility, cationic nature, and availability of modifiable functional groups. Some of the popular applications of chitosan nanomaterials include the preparation of bio-sensors, wound healing and wound dressing, gene delivery, and bone tissue engineering [72,73,74,75,76]. Additionally, chitosan conjugated folic acid was used for the synthesis of ZnS quantum dots, which are formulated into nanocarriers with the potential for suicide gene therapy [77], and chitosan-ZnS-FA nanoparticles were synthesized as a potential anticancer therapeutic agent [78]. Although the use of chitosan as an antimicrobial is extensively reviewed, more focused analysis on chitosan nanoparticles towards foodborne pathogens (FBPs) is scarce. Furthermore, the use of nanochitosan-based materials for cancer photothermal therapy is an emerging research field that is not broadly conversed. Hence, in this part of the review, the use of nanochitosan and its derivatives in controlling foodborne pathogens and in cancer photothermal therapy will be discussed in detail.

### 3.1. Chitosan Nanoparticles for Foodborne Pathogens

According to CDC estimates, every year in the United States, around 48 million cases of foodborne illnesses can occur together with 128,000 hospitalizations and 3000 deaths [79]. Additionally, as estimated by WHO, foodborne pathogens can cause 600 million cases of foodborne illnesses and 420,000 deaths in the globe every year, where 30% of the deaths are accounted for the children under five years of age [80]. Given these facts, in recent years, there has been an increased interest among researchers to discover novel antimicrobial agents against foodborne pathogens, and the use of chitosan-based nanoparticles to control foodborne pathogens is discussed in this section. 

#### 3.1.1. Direct Use of Chitosan Nanoparticles 

The analysis of the antimicrobial activity of low and high molecular weight chitosan nanoparticles (CSNPs) formulated using sodium sulfate or tripolyphosphate cross-linkers with or without sonication at different energy levels have shown to be effective against *Escherichia coli* O15:H7 [81]. CSNPs synthesized by ionic gelation to be used as an edible coating on grapes was reported to inhibit certain FBPs [82]. A vegetable wash prepared with CSNPs mixed with 1% citric acid has resulted in an increased reduction of the bacteria load on lettuce under simulated conditions compared to commercial formulations [83]. Mohammadi et al. (2016) reported that CSNPs showed significantly higher antimicrobial effect against *E. coli* compared to microparticles, and they have tested with different molecular weight chitosan formulations [84] (Table 2).

#### 3.1.2. Chitosan Nanoparticles with Essential Oils

CSNPs, combined with plant essential oils, have been widely tested against FBPs. CSNPs have shown better activity compared to nanocapsules of chitosan with or without lime essential oil [85]. Bio-nanofilms formulated with CSNPs, and fish gelatin with oregano essential oil (OEO) have shown an antimicrobial effect in a concentration-dependent manner [86]. The biofilms without OEO had not indicated antimicrobial effect, possibly due to lack of diffusion of CSNPs from film to agar plates [86]. CSNPs loaded with *Cyperus articulatus* Essential oil was reported to inhibit *Escherichia coli* and *Staphylococcus aureus* at much lower concentrations compared to individual components [87] (Figure 4). Nanoencapsulation of rosemary extract in chitosan and ɣ-poly glutamic acid at varying chitosan concentrations (0.1–1.8 mg/mL) was shown to have an improved antimicrobial effect with increasing chitosan concentration [88]. Rosemary essential oil (REO) encapsulated in CS-Benzoic acid nanogel (REO-CS-BA) was reported to have a higher inhibitory effect on *S. aureus* compared to free REO [89]. Additionally, starch-carboxy methylcellulose emulsion films formed by incorporating REO-CS-BA were shown to retain a certain level of antimicrobial properties [89]. CSNPs loaded with Cardamom essential oil (CEO) were able to inhibit multidrug-resistant pathogens related to food poisoning, and the effect was higher for the CEO loaded CSNPs compared to empty CSNPs [90]. Additionally, clove essential oil containing CSNPs has shown antimicrobial effect against four different FBPs with a good retention rate for the essential oil [91] (Table 3).

#### 3.1.3. Nanochitosan with Other Naturally Occurring Antimicrobials

Incorporation of CSNPs into edible films prepared for strawberry preservation together with 1% chitosan, glycerol, and ethanolic extract of propolis (EEP) has shown an inhibitory effect against FBPs in a concentration-dependent manner [92]. From these edible films, the higher effect was observed for formulation with 10% EEP, and the details for this film are recorded in Table 4. Chitosan nanofibers incorporated with Ag nanoparticles (AgNPs) synthesized using *L. salicaria* (purple loosestrife) medicinal plant extract has shown to be effective in the controlled release of AgNPs and inhibiting bacterial growth [93]. The effectiveness of CSNPs and CS-nisin nanoparticles in reducing the microbial growth in orange juice was analyzed using four foodborne pathogens. Here the reduction of microbial growth after inoculating the juice with each pathogen at initial concentrations around 8 CFU/mL was assayed and found to have a synergistic antimicrobial effect between CSNPs and nisin [94]. Additionally, encapsulation of nisin into chitosan or chitosan-monomethyl fumaric acid (CM) nanoparticles have shown to enhance the antibacterial effect of nisin against some foodborne pathogens [95]. The log reduction of growth by CM-nisin (starting from 7.5 log CFU/mL), which showed the highest effect, is recorded in Table 4. Chitosan coated liposome nanoparticles (chitosomes) loaded with nisin was shown to be effective against three common foodborne pathogens, and chitosan and nisin could be acting synergistically by reducing microbial nutrient uptake and inducing pore formation in the cell membrane [96]. Edible films made with chitosan-ZnO nanocomposite and cellulose using a sol-gel method by incorporating either nisin [97] or monolaurin [98] were found to inhibit the growth of *L. monocytogenes* on ultrafiltered cheese.

In another study, CSNPs loaded with the cell-free supernatants of lactic acid bacterial (LAB) cultures were used to test the antimicrobial effect of foodborne bacterial and fungal strains [99]. From the tested LAB cultures, *Lactobacillus helveticus,* which showed the highest antimicrobial activity, was used to analyze minimum inhibitory concentration together with CSNPs (Table 4). Tripolymeric nanoparticles containing chitosan, alginate, and pluronic F68 loaded with nisin were reported to have an enhanced effect compared to nisin itself, in inhibiting pathogenic microbes known to contaminate food [100]. A variety of formulations consisting of chitosan, with or without CSNPs, propolis nanoparticles, and propolis, was effective in controlling *Aspergillus* growth and aflatoxin production [101], and the values for the best formulation (40% nanoparticles) is recorded in Table 4. Additionally, Wang et al. (2015) suggested that nanoformulations of curcumin encapsulated with chitosan can be an effective way of improving sonodynamic antimicrobial properties of curcumin towards bacteria, including FBPs [102]. Nanoparticles of chitosan and natural cationic peptide protamine (CP) had demonstrated a higher inhibitory effect against *E. coli* and less effect on *B. cereus* [103]. In this study, bulk chitosan had elicited a higher level of inhibition against *B. cereus* compared to all the nanoparticle formulations, except when the protamine concentration was raised to 500 µg/mL. Although *B. cereus* was categorized as probiotic bacteria in this study, it is also known to cause food poisoning. pH-sensitive nanoparticles with Poly(D,L-lactide-co-glycolide) (PLGA) and chitosan encapsulating the natural antimicrobial trans-cinnamaldehyde (TCIN) has shown promising results for controlling FBPs, and the presence of chitosan in this delivery system was shown to have an enhanced effect against tested FBPs [104] (Table 4).

#### 3.1.4. Other Potential Applications of Chitosan Nanoparticles Related to Foodborne Pathogens

*Campylobacter jejuni* is a major source of food poisoning in the poultry industry. The hemolysin co-regulated protein (hcp) is believed to play a vital role related to the virulence of this bacteria as a component of the bacterial type VI secretion system, which is a phage like puncturing device [105]. 

A potential vaccine candidate against *C. jejuni* colonization in chicken was developed by entrapping recombinant hcp in chitosan-sodium tripolyphosphate nanoparticles, and the mucosal administration was found to have a significant effect [105]. Glucose oxidase (GOx) immobilized on CSNPs via covalent attachment (CA), enzyme coating (EC), enzyme precipitation coating (EPC), and magnetic nanoparticle-incorporated EPC methods were tested for the inhibition of *S. aureus* by enzymatically produced H_2_O_2_ in the presence of glucose. From these formulations, EPC and magnetic nanoparticle-incorporated enzyme-nanoparticles demonstrated good inhibition of the bacteria in both suspension cultures and biofilms [106].

Bulk chitosan-based materials with and without plant extracts were shown to have an antiviral effect against foodborne viruses (Murine Norovirus, MS2 phage, Feline calicivirus) [107]. Additionally, the use of chitosan-based material to treat human norovirus, the number one viral foodborne pathogen, has been tested [108]. But we didn’t come across any reports on nanochitosan against foodborne viruses. This shows a possible avenue for future research with nanochitosan against these viruses.

These examples show the direct use of nano-chitosan based materials or their usage as carrier molecules to improve the antimicrobial effect against common foodborne pathogens. The research has demonstrated that combined products have a higher impact compared to CSNPs by itself, indicating that the combination of different antimicrobial agents with chitosan can be more productive in controlling foodborne microbes.

#### 3.1.5. Possible Factors Affecting the Antimicrobial Effect of Chitosan

Chitosan-based antimicrobial activity is known to occur by increasing the permeability of the microbial cell membrane, resulting in the leakage of the cellular materials and cell death. The nanochitosan may have an increased effect due to the higher level of interaction with the microbial cell membrane owing to the large surface area and charge density [82]. The mode of action of chitosan as an antimicrobial agent and the factors that affect the antimicrobial activity of micro and nanochitosan are extensively reviewed in the literature [109,110,111]. Additionally, a study by Paomephan et al. has revealed that the antimicrobial effect of CSNPs against *E. coli* is mainly dependent on the surface area of the particles and not due to individual particle size or the molecular weight [83]. They have demonstrated that the CSNPs with similar size and zeta potential values elicited a similar inhibitory effect against *E. coli* and as the smaller size particles have a higher surface area, those were found to have a better effect [83]. It is possible that the larger surface area combined with higher charge density on smaller particles provides a higher level of interaction with the microbial surface resulting in superior inhibition. In contrast, when tested with *S.* Typhimurium, larger CSNPs have shown a greater effect, possibly due to the varied cellular envelop composition compared to *E. coli* [83]. Additionally, some studies have shown a relationship between molecular weight and inhibitory effects of chitosan [110]. Hence, more studies are needed to unravel the detailed mechanism of action of CSNPs against different bacterial strains, and it may vary according to the bacterial strain and their cell membrane/envelop composition. 

### 3.2. Role of Chitosan in Cancer Photothermal Therapy

During recent years, cancer has become a major health crisis throughout the globe costing thousands of lives each year. According to the World Health Organization (WHO), every sixth death in the world is caused by cancer [112]. The existing treatments, such as chemotherapy and radiation therapy, can be challenging due to harmful effects exerted on adjacent healthy tissues. Chemotherapeutic agents can affect the noncancerous cells that are dividing rapidly. Additionally, the other issues with chemotherapeutics include systemic accumulation, developing drug resistance, and lower effective concentrations at the target sites, leading to undesirable outcomes in cancer patients. The high doses of radiation used in radiotherapy are known to heighten the invasive properties of cancer cells, and certain cancer types have developed resistance to the radiation. Hence, targeted thermal treatments such as photothermal therapy have gained the attention of scientists due to its advantages such as cost-effectiveness, reduced side effects, and noninvasive nature [112]. Nanoparticles containing photothermal agents introduced into tumor sites can be induced by near-infrared (NIR) light in the wavelength range at first (700–980 nm) or second (1000–1400 nm) biological window [113]. Living tissues and biological substances minimally absorb light in the NIR window leading to lower phototoxicity and have deeper tissue penetration compared to UV or visible light, making it the preferred light source for cancer therapy [114]. Various types of photothermal agents such as carbon-based compounds (graphene oxide, carbon quantum dots), inorganic nanomaterials (gold, silver, and copper nanoparticles), and NIR sensitive dyes have been extensively studied in the literature. However, their application as photothermal agents has been limited due to characteristics such as poor internalization, low stability, high toxicity, and lower biocompatibility. Chitosan and its water-soluble derivatives have been successfully used to overcome these issues due to its cationic nature, biocompatibility, and swelling properties. The chemical structure of chitosan has partial similarity to hyaluronic acid, which acts as a ligand for CD44 receptors that are highly expressed on several cancer cells. For example, a nanocarrier system formulated with chitosan has shown impressive anticancer effect against breast cancer spheroids overexpressing CD44 receptors showing its potential as a targeting moiety [115].

Although there are many reviews providing an overview related to the use of nanochitosan in cancer therapy, the usage of chitosan-based nanomaterials related to photothermal cancer therapy is scarce.

Hence this section is added to describe the role of chitosan-based nanomaterials in cancer photothermal therapy. 

#### 3.2.1. Applications of Nanochitosan in Photothermal Therapy

Magnetic graphene oxide–chitosan (CS)-sodium alginate (SA) nanocomposites were prepared by depositing layers of oppositely charged CS and SA one after the other, driven by electrostatic forces. These polymeric layers have assisted in avoiding aggregation of the nanocomposite films and appeared to have favorable properties as a therapeutic agent [116]. A charge reversible, positively charged chitosan crosslinked-single walled carbon nanotube core embedded in negatively charged polyethylene glycol was synthesized with the ability to target mitochondria [117] (Table 5). In another chemo-PTT therapeutic, chitosan coating of the PTT agent had lowered the cytotoxicity of the compound, indicated by much less cell cycle defects compared to uncoated product [118]. Chitosan was used to coat silver nanoparticles as a PTT agent, to provide a protective layer to avoid the aggregation and to make it biocompatible under physiological conditions [119]. Use of chitosan in a drug delivery system consisting of gene silencing siRNA and gold nanorods had shown remarkable stability, low cytotoxicity, good cellular uptake, and escaped from endosomal/lysosomal structures [120]. A multifunctional scaffold coated with nano-hydroxyapatite and graphene oxide with the ability to kill osteosarcoma cells and promote osteogenesis was developed with chitosan due to its high biocompatibility and hemostatic effect [121]. During the synthesis of another multifunctional chemo-photo therapeutic agent, chitosan was utilized as a reducing agent for graphene oxide and helped to stabilize reduced graphene oxide by forming a layer around it [122]. Additionally, chitosan worked as a shell to carry DOX and IR820 dye in different ratios and allowed the slow release of DOX after internalization due to its “pH-dependent dissolution and swelling properties” [122] (Figure 5). Similarly, the pH based swelling of chitosan was utilized in a targeted drug delivery system containing hollow mesoporous silica nanoparticles to deliver therapeutics without leakage [123].

Chitosan Used with Metallic Substances

Chitosan-molybdenum disulfide nanosheets decorated with tantalum oxide has demonstrated higher photostability and cytotoxic effects against breast cancer cells compared to bare MoS_2_ particles, where chitosan served as the surface matrix allowing the formation of the final product [124]. Chitosan was used in the synthesis of a nanocomposite with the combined ability of tetra-modal imaging and PTT, using upconversion nanoparticles (UCNPs) and Ag_2_Se. Here chitosan was incorporated to impart stability, water dispersibility, and biocompatibility to UCNPs while providing a platform for Ag_2_Se to grow [125]. Additionally, iron crosslinked chitosan-based complexes were used as the precursors during the green, hydrothermal synthesis of a biocompatible therapeutic agent [126] (Table 5).

#### 3.2.2. Compounds Synthesized with Chitosan-Derivatives

Multifunctional nanoparticles with potential photothermal, photodynamic, chemotherapeutic, and imaging properties were developed using chitosan derivative based-AZA-boron dipyrrolide (AZA-BODIPY) nanoshell carrier loaded with the chemotherapeutic methotrexate (MTX) and was shown to be effective at a lower dosage compared to the MTX itself [127]. Gold nanorods coated with layers of hydroxyethyl chitosan and hyaluronic acid were able to create a pH-sensitive, surface charge reversible, tumor-targeted drug delivery system, with the potential to act as chemo-photothermal agent [128] (Figure 6). Additionally, water-soluble hydroxyethyl chitosan provides modifiable functional groups, allowing the attachment of other compounds/drugs, and being a cationic polymer; it permits easy uptake through the negatively charged cell membrane [129]. Quaternized chitosan was used as a biotemplate and stabilizing agent during the green synthesis of CuS nanoparticles for photothermal based cancer therapy [130]. Chitosan coating of gold nanorods is shown to eliminate the cytotoxic effects of cetyltrimethylammonium bromide (CTAB) that is usually absorbed on to the surface of the nanorods after the synthesis process. Chitosan also helped to stabilize and avoid aggregation of the nanorods after replacing CTAB [131]. Functionalization of reduced graphene oxide nanosheets with carboxymethyl chitosan has enabled better distribution of the compound in a 3D hydrogel and enhanced the absorption of NIR light [132]. Chauhan et al. (2018) functionalized the poly (lactic-co-glycolic acid) nanoparticles with glycol chitosan due to its water solubility and “enhanced permeability and retention effect,” leading to the production of plasmonic carbon nanohybrids [133]. Thiol chitosan was used to coat gold nanoshells in a multifunctional theranostic drug, allowing the loading of the anticancer drug paclitaxel, and increasing biocompatibility [134]. During the synthesis of gold nanoshells, glycol chitosan was used to functionalize PLGA nanoparticles, providing a cationic surface with amine functional groups to drive the in situ reduction of gold [135]. A concentration-dependent effect of glycol chitosan was observed during the coating of PLGA, as higher concentrations lead to the formation of micro-size polypod structures with the deposition of gold. These polypod structures formed with 0.25% glycol chitosan were claimed to be not appropriate for photothermal studies owing to their aggregated nature and larger size. But, continuous and branched structures were obtained with 0.025% glycol chitosan and had demonstrated good photothermal effect [135]. Water-soluble chitosan functionalized pluronic nanogel was used to load photo-therapeutic agents as it had shown better tumor targeting and accumulation as a carrier during previous studies [136,137]. 2,3-dimethylmaleic anhydride (DMA)-modified chitosan conjugated to oligosaccharide-block-poly(ethylene glycol) incorporated into a nanoplatform could serve as a charge-reversible, antifouling polymer [138]. Upon exposure of this compound to acidic pH at tumor sites, cleavage of amide bonds between chitosan and DMA led to the removal of this segment due to electrostatic repulsion and facilitated internalization of the therapeutic nanoparticles. Sonication of single-walled carbon nanotubes with water-soluble chitosan has allowed the suspension of the nanotubes and provided amine groups to attach CD133 monoclonal antibody-phycoerythrin via carbodiimide chemistry in developing a targeted therapeutic agent [139]. Chitosan coated nano-CpG (cytosine-guanine) platform containing hollow CuS nanoparticles was synthesized by Guo et al. for combined photo-immunotherapy [140]. Upon irradiation with NIR light, this system transformed into chitosan-CpG nanocomplexes and released single CuS crystals. These chitosan-CpG complexes have increased tumor retention and enhanced uptake of CpG via plasmacytoid dendritic cells compared to free CpG [140] (Table 6).

In addition, chitosan has been used to create thermoresponsive carrier systems for chemotherapeutics together with NIR responsive material, where the primary focus was NIR based drug release and not the therapeutic effect of light. Positively charged, water-soluble chitosan derivative was conjugated onto chemically reduced graphene oxide to gain stability and avoid aggregation when incorporated into a reversible NIR responsive nanogel designed for the release of DOX [141]. Similar thermal and pH-sensitive DOX carrier was developed using alkyl chain functionalized chitosan, and a Schiff base formed between primary amino groups of chitosan and ketone groups of DOX has imparted pH sensitivity to the system [142] (Table 6).

#### 3.2.3. The Photothermal Effect

The therapeutic effect of photothermal agents arises from their ability to convert NIR light to heat energy creating local hyperthermia [143]. In noble-metal nanoparticles, the photothermal effect (light to heat conversion) arises when the incoming light coincides with the oscillation frequency of the conduction electrons, generating localized surface plasmon resonance (LSPR) [119]. Boca et al. claimed their article to be the first report of using chitosan to coat nanoparticles to be used as a photothermal agent. In carbon-based material such as carbon nanotubes, the heat generated by the vibrational modes of the lattice arises from the optical transitions causing excitation and subsequent relaxation [144]. In transition metal nanoparticles like CuS, d-d transition of metal ions drives the absorption of near-infrared light [145]. During PTT, when the temperature rises up to 40–45 °C around the tumor site, it causes many changes in the cancer cells, such as protein denaturation, swelling of mitochondrial membrane, and breakage of cellular membrane leading to cell death. But noncancerous cells were found to be immune to such changes up to 1 h of similar light treatment, making PTT a targeted and minimally invasive therapy [145,146]. The researchers revealed that the PTT dependent cancer cell death could occur from either apoptosis or necrosis, and the more desirable apoptotic pathway, which depresses inflammatory response, can be induced in cells by controlling the NIR light intensity and the exposure time [147].

## 4. Toxicity of Nanochitosan

According to the literature, concentration dependant toxicity of nanochitosan was observed with certain plants. Targhi et al. have reported potential phytotoxicity of both nano-size and bulk chitosan on pepper plants (*Capsicum annuum* L.) at higher concentrations of 5 to 20 mg/L and 100 mg/L, respectively, while eliciting a growth-promoting effect at lower concentrations [148]. Additionally, nanochitosan around 20 nm was shown to have toxic effects on germination and seedling growth of broad beans after seed priming at 0.05% and 0.1% concentrations [149]. At the same time, an increase in defensive phenols and antioxidant enzymes was seen at a lower 0.05% concentration for broad beans [149]. On the other hand, the application of nanochitosan at 50 ppm concentration was shown to support microbial activity and soil health [70]. Hence, the nano-chitosan concentration has to be controlled to obtain positive effects while minimizing the toxic side effects on plants.

Additionally, the use of chitosan-silver nanoparticles (Ch-AgNPs) to control the malaria vector *anopheles stephensi* was reported by Murugan et al., 2016 [150]. The predation efficiency of zebrafishes for I and II instar larvae of the malaria vector was shown to increase in the presence of Ch-AgNPs (1 ppm) under the aquatic environment. Whereas at the levels of 8 to 10 ppm, harmful effects were observed for a non-target crab species *P. hydrodromous*. Furthermore, the possibility of using chitosan and nano-chitosan isolated from shrimp and a fungal source as a food ingredient was studied using rat and brine shrimp bioassays [151]. Feeding of both forms of chitosan to rats at 100 to 200 mg/Kg did not show any adverse effects on the liver, kidneys, and stomach tissues as well as on blood biochemical and oxidative stress levels. Other studies have also indicated that different types of chitosan had no toxic effects on the liver and kidneys. During brine shrimp assay, chitosan and nano-chitosan from 5000 to 15,000 ppm were shown to be nontoxic and slight toxicity was observed at the concentrations above that [151]. Additionally, exposure of zebrafish embryo to chitosan nanoparticles was shown to cause a dose-dependent lower hatching rate and higher motility. The smaller size (200 nm) nanoparticles used in zebrafish studies were found to be more toxic compared to larger size nanoparticles (340 nm), and the observed toxicity was shown to be a result of increased apoptosis, oxidative stress, and production of heat shock protein 70 [152]. Therefore, it is clear that the species-dependent toxicity can exist with the use of nanochitosan-based material, which may be minimized by adjusting the dosage and size of the nanoparticles. Due to the small size and the ability to penetrate through biological membranes, nanoparticles pose the risk of entering biological systems. Therefore, the concentration and size dependent properties of nanochitosan and its derivatives should be properly analyzed before the usage to avoid any toxic effects.

## 5. Future Directions and Concluding Remarks

The usage of nontoxic, biodegradable polymer systems like chitosan for the progress of the agricultural and biomedical fields is beneficial for the society and the ecosystem. Among the many uses of nanochitosan in the agricultural sector, its involvement in alleviating abiotic stresses and application in water purification for agricultural purposes can maximize the land and water usage for crop production. Compared with conventional materials, renewable chitosan nanoparticles used as a bioflocculant and a heavy metal adsorbent demonstrate better or compatible performance in industrial or agricultural wastewater treatments. In the face of climate change, the well-treated non-conventional water (e.g., industrial wastewater) can be a viable option for crop irrigation to enhance food security for the increasing global population. Even though nano-chitosan has been widely studied related to its uses as a fertilizer, herbicide, insecticide, and a carrier system, additional research is needed to exploit its capabilities in abiotic stress management. Notably, in combating heat and heavy metal stress, studies with bulk chitosan have shown promising results, but there is a lack of reports on the usage of nanochitosan in these areas. Due to the chelating properties of chitosan, it can also be a useful soil conditioner to complex toxic metals in polluted soil. Additionally, it is possible to adapt nano-chitosan based heavy metal remediation methods used in other areas such as water purification and biomedical treatments to suit applications in soil treatments. Moreover, there are only a few studies on the use of chitosan derivatives to reduce abiotic stress in plants, even in the bulk form. Therefore, the use of nano-chitosan derivatives can be a novel area with a high potential to combat abiotic stress in plants that can be explored in the future.

When considering antimicrobial studies, chitosan is used together with a variety of other natural antimicrobials such as essential oils, propolis from beehives, curcumin, and peptides to control FBPs. It is also clear that the combined effect of chitosan with other antimicrobials is higher in most of the reported studies, displaying the synergistic antimicrobial effect of chitosan. However, the use of chitosan-based nanomaterials for treating viral foodborne pathogens is not well studied. Hence, this can be a future research avenue related to the antimicrobial activity of chitosan. Additionally, nanochitosan and its derivatives have been extensively used in the production of photothermal agents to treat cancer. Even though chitosan is not a photothermal agent by itself, its biocompatible and swelling properties have assisted in formulating therapeutics with required characteristics. For example, chitosan and its derivatives were able to impart greater stability, dispersibility, increased tumor retention and helped to alleviate cytotoxicity of the nanotherapeutic agents. Liu et al., 2019 reported that the incorporation of carboxymethyl chitosan on graphene oxide nanosheets had enhanced the absorption of NIR light by the compound [132]. Chitosan was used as a reducing agent for graphene oxide-based therapeutics and gold and as a precursor for the synthesis of other therapeutics [122,126,135]. Additionally, chitosan derivatives such as thiol chitosan, hydroxyethyl chitosan, and glycol chitosan have widely been used in nanoformulations to enable cellular internalization and loading of other drugs to form multifunctional therapeutics. Furthermore, pH-sensitive drug delivery systems were developed using chitosan due to its ability to undergo reversible protonation and deprotonation of amine groups inducing pH-controlled drug release in the acidic tumor microenvironment [153]. Chitosan is also known to contain inherent anticancer properties, based on its ability to induce the production of Tissue Necrotic Factor-α via monocytes, further enhancing its value in cancer therapy [154]. Owing to its nontoxic, biocompatible, and anticancer nature, the use of nano-chitosan and derivatives in noninvasive cancer therapies such as PTT show great potential for future cancer treatments.

Despite the many records related to applications of nanochitosan, still, there is more work to be done to bring these up to field trials and clinical usage. Although chitosan is known to be nontoxic by nature, when formulated into nanoparticles, dosage-dependent issues can arise due to its enhanced ability to accumulate in the water, soil, and biological systems compared to the bulk material. Additionally, other compounds incorporated with chitosan, such as metals, can pose an environmental and health risk during the usage. Therefore, more research needs to be conducted related to the toxicity of these formulations before the final stages of application. Furthermore, a better understanding of the mechanism of action of nanomaterials related to agricultural and biomedical fields can shed light on proper usage of those with minimum damage to the environment, while obtaining the highest benefits for human beings.

## Figures and Tables

**Figure 1 nanomaterials-10-01903-f001:**
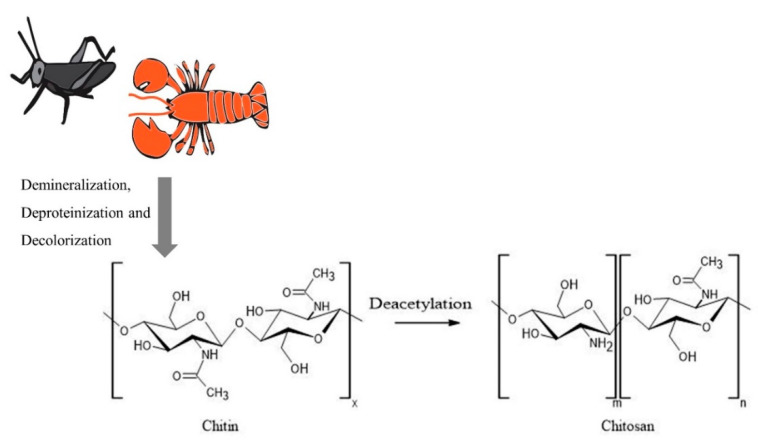
Process of chitosan production starting with different sources. The figure was created using ACD/ChemSketch and Adobe Illustrator 2020.

**Figure 2 nanomaterials-10-01903-f002:**
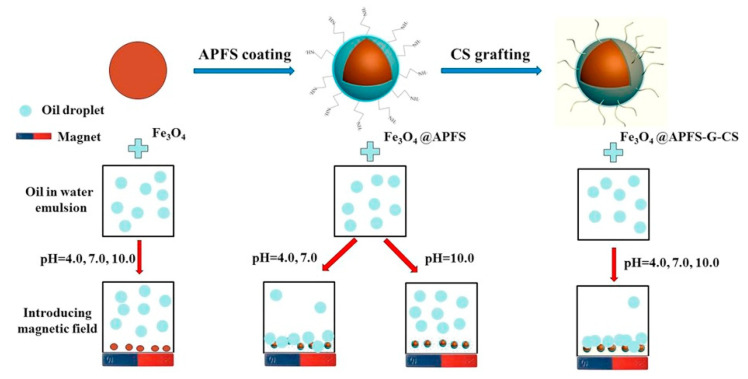
Schematic representation of the efficient removal of oil droplets from emulsified oil wastewater with aminopropyl-functionalized silica (APFS) coated, chitosan-grafted magnetic nanoparticles. An enhanced demulsification effect was observed after grafting with chitosan under all the tested pH conditions. Reproduced with permission from Reference [26] Copyright © 2017 Elsevier.

**Figure 3 nanomaterials-10-01903-f003:**
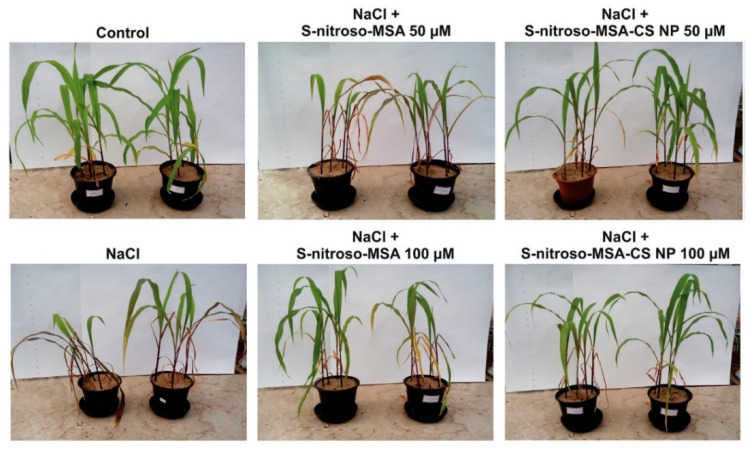
Comparison of maize plants under salt stress treated with free S-nitroso-mercaptosuccinic acid (MSA) to the plants treated with chitosan nanoparticles encapsulating S-nitroso-MSA at 50 or 100 µM concentration. It shows that the treatment with S-nitroso-MSA-Chitosan nanoparticles at both concentrations was effective in relieving the visible effects (necrosis) of salt stress in the plants compared to free S-nitroso-MSA, which has shown to be effective only at 100 µM concentration. Control plants were treated only with distilled water or salt without any treatment. Reproduced with permission from Reference [51] Copyright© 2016 Elsevier.

**Figure 4 nanomaterials-10-01903-f004:**
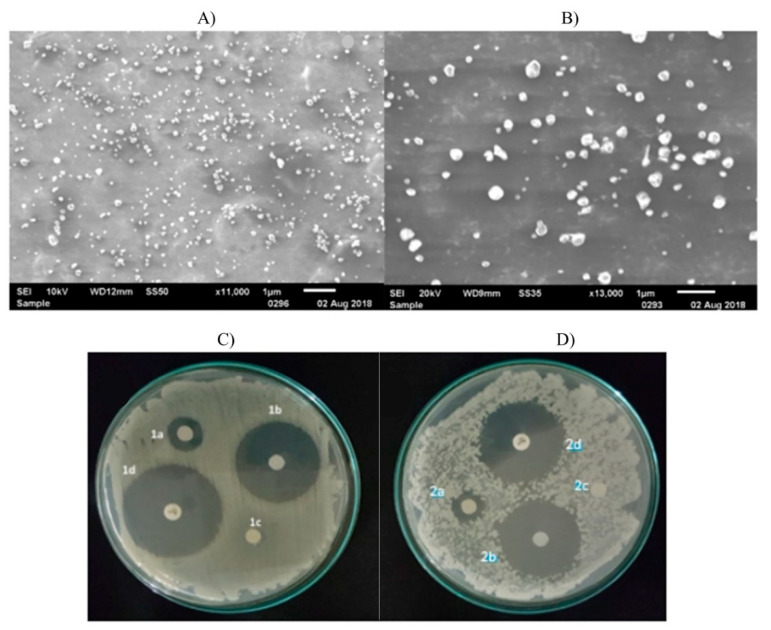
(**A**) Scanning Electron Microscopy images showing CSNPs and (**B**) *Cyperus articulatus* Essential oil (CPEO) loaded CSNPs synthesized with 1:0.25 weight ratio of chitosan to essential oil. (**C**) Initial analysis of the antimicrobial effect of CSNPs-CPEO using disc diffusion method with *S. aureus* and (**D**) *E. coli* together with control experiments. In figures (**C**) and (**D**), the inhibition zone for free essential oil is indicated as 1a and 2a, respectively. Similarly, 1b and 2b show CSNPs-CPEO inhibition zone, 1c and 2c: empty disc used as the negative control, and 1d and 2d: Ciprofloxacin used as the positive control. Reproduced with permission from Reference [87] Copyright © 2019 Elsevier.

**Figure 5 nanomaterials-10-01903-f005:**
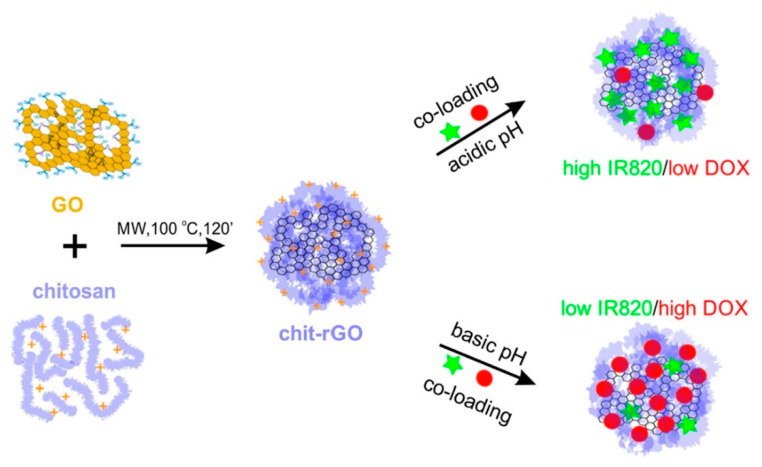
The use of polymeric chitosan in the formulation of a multifunctional chemo-phototherapeutic agent. Chitosan was used as a reducing and stabilizing agent for reduced graphene oxide nanoflakes and also served as a shell to entrap DOX and IR820 dye at different ratios. Reproduced with permission from Reference [122] Copyright © 2019 Elsevier.

**Figure 6 nanomaterials-10-01903-f006:**
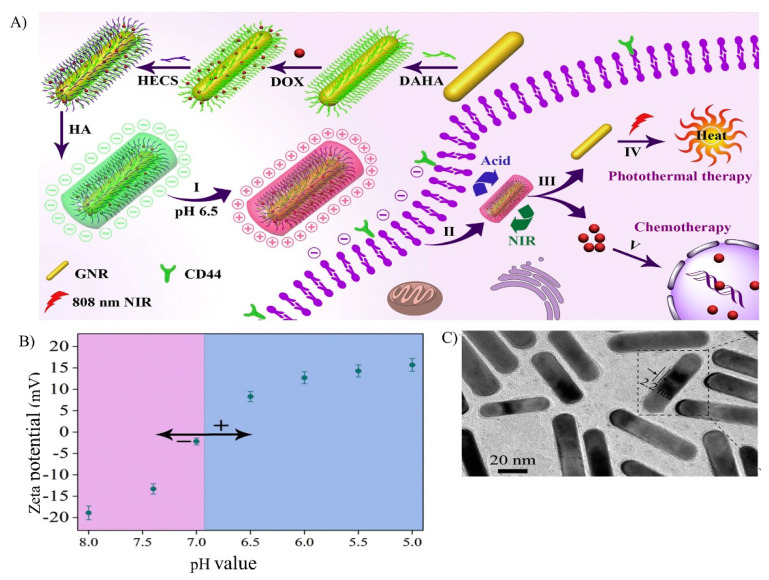
(**A**) Preparation of hydroxyethyl chitosan derivative (HECS) and hyaluronic acid (HA) coated gold nanorods functionalized with aldehyde/catechol- hyaluronic acid (DAHA) and loaded with DOX (gold nanorod-DAHA^DOX^-HECS-HA), as a charge reversal, pH/NIR responsive therapeutic agent. (**B**) The charge-reversible nature of gold nanorod-DAHA^DOX^-HECS-HA was analyzed by measuring the zeta potentials at different pH values. When the pH of the media was reduced from 8.0 to 5.0, the zeta potential increased from −18.9 to 15.7 mV, indicating charge reversal. (**C**) TEM image of gold nanorod-DAHA^DOX^-HECS-HA showing the shell thickness of 2.2 nm. Reproduced with permission from Reference [128] Copyright © 2019 Elsevier.

**Table 1 nanomaterials-10-01903-t001:** Applications of chitosan nanoparticles in drinking water purification and agricultural/industrial wastewater treatment.

Nanoparticles	Targeted Pollutants	Effectiveness and/or Efficiency	Reference
Nanochitosan	Pb(II) in water	Adsorption capacity: 32.26 mg/g at pH 6	[30]
Magnetic chitosan nanoparticles	Pb(II) and Cd(II) in wastewater	Adsorption capacity: 79.24 mg/g for Pb(II) and 36.42 mg/g for Cd(II)	[23]
Magnetic chitosan polyelectrolyte nanoparticles	Cd(II) in industrial wastewater	97.5 removal from the original 100 mg/L concentration	[24]
Chitosan nanoparticle	Cr(III) in tannery wastewater	70% removal of chromium in 24 h	[31]
chitosan magnetite nanoparticles	Cr(VI) in wastewater	75–88% removal from the standard 500 mg/L K_2_Cr_2_O_7_ solution	[32]
Magnetic chitosan nanoparticles	Cr(VI) in wastewater	Adsorption capacity: 58.14 mg/g at pH 3.0	[33]
chitosan-stabilized Fe/Cu bimetallic nanoparticles	Cr(VI) in different types of water	Removal efficiency: 90% (river water), 85%(tannery water), and 80% (smelting water)	[34]
Chitosan-/PVA-coated magnetic nanoparticles	Cu(II) in wastewater	Adsorption capacity: up to 500 mg/g at pH 5.0	[35]
Chitosan gel nanoparticles	Cu(II) in wastewater	Adsorption capacity: 78–112 mg/L	[36]
Chitosan magnetite nanoparticles	Heavy metals in the water part of the sludge	Adsorption: 20–50% more heavy metals than magnetite	[37]
Chitosan nanoparticles	Eu(III) in water	Adsorption capacity: 114 mg/g, >30 times compared to crab shell particles	[22]
Chitin nanocrystals	Ag(I) in water	27% removal from the original 107.8 mg/L concentration	[38]
Magnetic chitosan nanoparticles	Azo dyes in wastewater	94–96% removal at pH 6.0 in 1 h	[39]
Magnetic chitosan nanoparticles	Dyes in wastewater	Adsorption capacity: 82.2 mg/g for removing Bromothymol Blue	[40]
Chitosan-silica nanoparticles with immobilized Cu(II) ions	1,1–dimethyl hydrazine in wastewater	100% degradation of 1,1–dimethyl hydrazine in 10 min	[41]
Chitosan modified multi-wall carbon nanotubes	Phosphate in wastewater	Adsorption capacity: 36.1 mg P/g, and 94–98% of the original efficiency after 5 cycles	[42]
Enzymatic chitosan nanoparticles	Phenols in wastewater	Higher thermostability than free enzyme and same activity	[43]
Highly deacetylated chitosan nanoparticles	Diclofenac and carbamazepine in wastewater	Adsorption capacity: up to 351.8 mg g^−1^ for diclofenac	[44]
Chitosan−silver Nanoparticles	Bacteria in drinking water	99.99% removal of bacteria in 15 min, and complete removal in 8 h	[45]
Chitosan-coated silver nanoparticles	Various toxic contaminants	Inhibition of biofilm formation	[46]
2(5H)–furanone loaded chitosan nanoparticles	COD* and color in Rice mill wastewater	Better foulant rejection, better removal of COD, and color	[27]
chitosan-doped MIL-100(Fe) nanoparticles	Bacteria in wastewater	higher biofouling resistance of 85% comparedto the original 51%	[47]
Silver-loaded chitosan nanoparticles	Foulants on hollow fiber membranes	Optimal rejection of 89.27 and 86.04% for Reactive Black 5 and Reactive Orange 16	[48]
O-carboxymethyl chitosan-Fe_3_O_4_ nanoparticles	Foulants on membranes	Achieving the lowest irreversible fouling resistance of 4.2% at 0.05 wt.%	[28]
chitosan-grafted magnetic nanoparticles	Oil drops in emulsified wastewater	Best flocculation performance at pH 4.0, and reuse up to 7 times	[26]

COD: Chemical oxygen demand.

**Table 2 nanomaterials-10-01903-t002:** Direct uses of chitosan nanoparticles to control foodborne pathogens (MIC: minimum inhibitory concentration; MBC: minimum bactericidal concentration).

Compoxund	*Foodborne Pathogens*	Major Method of Analysis	Values from Analysis	Particle Size (nm)	Reference
CSNPs	*Escherichia coli* O157:H7	Log reduction (units not given)	0.4–9.7	<300	[81]
Edible coating of CSNPs on grapes	*Salmonella* spp.	MIC (g/L) vs. MBC (g/L)	3.0 vs. 6.0	128.3	[82]
	*E. coli*		3.0 vs. 3.0		
	*S. aureus*		2.0 vs. 6.0		
	*P. aeruginosa*		3.0 vs. 4.0		
	*L. monocytogenes*		3.0 vs. 6.0		
Vegetable wash with CSNPs and 1% citric acid	*E. coli*	Reduction of viable bacteria (log CFU/g)	1.63	352.7 ± 2.8 (most effective size)	[83]
	*S.* Typhimurium		1.16	865.9 ± 15.3 (most effective size)	
Low molecular weight CSNPs	*E. coli*	MIC(%*w/v*) vs. MBC (%*w/v*)	0.018 vs. 0.037	60 ± 5.48	[84]
Medium molecular weight CSNPs			0.037 vs. 0.075	78.50 ± 6.77	
Middle-viscous CSNPs (crab shell CS)			0.037 vs. 0.075	105.20 ± 8.58	

**Table 3 nanomaterials-10-01903-t003:** Use of chitosan nanoparticles with plant essential oils against foodborne pathogens. (MIC: minimum inhibitory concentration, MBC: minimum bactericidal concentration).

Compound	*Foodborne Pathogens*	Major Method of Analysis	Values	Particle Size (nm)	Reference
CSNPs- lime essential oil (LEO)	*S. aureus*	Minimum Inhibitory Volume (µL) for CSNPs-LEO:CSNPs	1.25:2.5	4.7 ± 1.2 (CSNPs) 6.1 ± 0.4 (CSNPs-LEO)	[85]
	*L. monocytogenes*		1.25:1.25		
	*S. dysenteriae*		1.25:1.25		
	*E. coli*		2.5:5		
CSnanocapsules(CSNC) – lime essential oil (LEO)	*S. aureus*	Minimum Inhibitory Volume (µL) for CSNC-LEO:CSNC	5:10	5.8 ± 1.6 (CSNC) 6.1 ± 0.6 (CSNC-LEO)	
	*L. monocytogenes*		5:20		
	*S. dysenteriae*		5:no inhibition		
	*E. coli*		10:no inhibition		
Fish gelatin/CSNPs-oregano essential oil bio-nanofilm	*S. aureus*	Agar diffusion Method (highest effect observed at 1.2 (% *w/v*) OEO)	26.33 ± 0.57	40–80	[86]
	*L. monocytogenes*		26.66 ± 1.52		
	*S. enteritidis*		30.33 ± 1.15		
	*E. coli*		33.00 ± 1.00		
CSNPs-*Cyperus articulatus* Essential oil (CPEO) (1: 0.25)	*E. coli*	MIC (mg/L) vs. MBC (mg/L)	5 vs. 10 (CSNP-CPEO) 40 vs. 80 (CSNPs) 10 vs. 20 (CPEO)	119 (CSNP-CPEO)	[87]
	*S. aureus*		10 vs. 15 (CSNP-CPEO) 80 vs.160 (CSNPs) 20 vs. 25 (CPEO)		
Rosemary extract loaded NPs with CS and ɣ-PGA	*B. subtilis*	Log reduction of growth in Barley tea (log CFU/mL)	More than 0.5–3.6	200–600	[88]
Rosemary essential oil encapsulated in CS-Benzoic acid nanogel	*S. aureus*	MIC (µg/mL)	40	Less than 100	[89]
Cardamom oil (CDEO) loaded CSNPs	*E. coli* (ESBL positive)	OD based micro-dilution broth assays	CDEO –CSNPs Maintained antimicrobial effect for 7 days against both pathogens. CSNPs alone was effective only for 48 h.	50–100	[90]
	*S. aureus* (Methicillin- resistant)				
Clove essential oil (CEO) loaded CSNPs	*L. monocytogenes*	Minimum inhibitory volume (µL) CEO-CSNPs: CEO: CSNPs	2:2:8	223–444	[91]
	*E. coli*		2:4:8		
	*S. aureus*		2:2:8		
	*S. typhi*		2:2:8		

**Table 4 nanomaterials-10-01903-t004:** Use of chitosan nanoparticles together with natural antimicrobial agents to control foodborne pathogens (MIC: minimum inhibitory concentration, MBC: minimum bactericidal concentration).

Compound	*Foodborne Pathogens*	Major Method of Analysis	Values	Particle Size (nm)	Reference
Edible film with CSNPs and 10% EEP	*E. coli*	CFU on Agar plates in 24 h vs. 48 h	0 vs. 5.67	28.42 ± 7.43 (for CSNPs)	[92]
	*L. monocytogenes*		0 vs. 43.33		
	*S. enteritidis*		11.33 vs. 13.33		
Chitosan nanofiber-AgNPs	*E. coli* O157:H7	Inhibition zone observed by Agar well diffusion method (mm)	14.54 ± 0.23	40 (nanofibers) 45–60 (AgNPs)	[93]
	*S. aureus*		17.62 ± 0.205		
Nisin-CSNPs (against pathogens inoculated in orange juice)	*S. aureus*	Log reduction of growth (CFU/mL) for Nisin-CSNPs:CSNPs	3.82 ± 0.03: 2.21 ± 0.01	147.93 ± 2.9 (Nisin-CSNPs) 64.34 ± 2.12 (CSNPs)	[94]
	*L. monocytogenes*		3.61 ± 0.05:2.15 ± 0.04		
	*E. coli O157:H7*		3.49 ± 0.01:2.03 ± 0.03		
	*S.* Typhimurium		2.88 ± 0.03:1.96 ± 0.01		
Nisin-chitosan-fumaric acid	*S. aureus*	Log reduction of growth in 24 h (CFU/mL)	3.43	207.93 ± 4.72	[95]
	*L. monocytogenes*		3.30		
	*E. coli* O157:H7		3.33		
Chitosomes with nisin	*S. aureus*	Minimum Inhibitory Concentration for nisin (µg/mL)	5	50–108	[96]
	*L. monocytogenes*		50		
	*Enterococcus faecalis*		200		
Edible Nisin loaded bilayer film with cellulose and chitosan-zinc oxide nanocomposite	*L. monocytogenes*	Log reduction of growth in UF cheese after 14 days (log CFU/g)	2.7 (500 ppm nisin film) 5 (1000 ppm nisin film)	Not given	[97]
Monolaurin incorporated nanostructured chitosan-zinc oxide-cellulose films	*L. monocytogenes*	Log reduction of growth in UF cheese after 14 days (log CFU/g)	2.4 (0.5% Monolaurin film) 2.3 (1% Monolaurin film)	Not given	[98]
Cell-free LAB culture supernatant loaded on CSNPs	*Staphylococcus sciuri*	Minimum Inhibitory Concentration (mg/mL)	46.7 ± 2.77	5–10 (size was reported only for natamycin control loaded with CSNPs)	[99]
	*Bacillus cereus*		43.3 ± 1.39		
	*Salmonella enterica*		40 ± 0.00		
	*Escherichia coli*		80 ± 0.00		
	*Pseudomonas aeruginosa*		80 ± 0.00		
	*Penicillium chrysogenum*		175 ± 0.00		
	*Candida parapsilosis*		550 ± 0.00		
Nisin loaded alginate-chitosan-pluronic F68 nanoparticles	*P. aeruginosa*	Absorbance of inoculated samples at 600 nm	Inhibited microbial growth at least 20 days in nutrient media and up to 6 months in tomato juice	208.2–831.9	[100]
	*S. enterica*				
	*E. aerogenes*				
CS+ 40% Propolis NPs	*Aspergillus flavus*	% mycelial inhibition: % germination inhibition: Aflatoxins (µg/L)	28.9:12.3:2.7	3.0 (CSNPs) 2.33 (PropolisNPs)	[101]
CS+ CSNPs			21.2:8.0:1.5		
CS+ Propolis extract			15.3:4.5:2.0		
CS+ 20% CSNPs+ 20% propolis Nps			30.4:1.3:2.8		
CS+ 20% CSNPs+ 20% propolis Nps+ Propolis extract			33.0:96.6:2.8		
CS+ 40% propolis Nps + Propolis extract			18.1:55.5:2.6		
CS+ 40% CSNPs + Propolis extract			19.0: 6.5:2.5		
CS-protamine nanoparticles	*E.coli*	MIC vs. MBC (µg/mL)	31.25 vs. 31.25–62.5	27.67–32.23	[103]
	*B. cereus*		31.25 to >250 vs. >250		
PLGA-chitosan-TCIN nanoparticles	*S.* Typhimurium	MIC (µg/mL) vs. MBC (µg/mL)	~16 vs. >64	277.3–295.0	[104]
	*S. aureus*		~16 vs. >64		

**Table 5 nanomaterials-10-01903-t005:** Applications of nanochitosan in cancer photothermal therapy.

Type of Chitosan Nano Polymer	Photothermal Agent	Tumor Models Used	Wavelength of Laser Source (nm)	Reference
Layer by layer modification with chitosan and sodium alginate	Graphene oxide	A549 human lung cancer cells	808, 1 W/cm^2^	[116]
Glutaldehyde-crosslinked chitosan layer	Single-walled carbon nanotubes	MB49 Murine bladder cancer cells	808, 2 W/cm^2^	[117]
Chitosan coating	Graphite carbon nanocages	CNE human nasopharyngeal cells and in vivo studies with tumor-bearing BALB/c nude mice	808, 0.25 W/cm^2^ co-irradiated with microwave radiation (2–10 W, 2450 MHz)	[118]
Chitosan coating	Silver nanotriangles	NCI-H460 human non-small lung cancer cells	800, 12–55 W/cm^2^	[119]
Chitosan coating	Gold nanorods	MDA-MB-231 human breast cancer cells and in vivo studies with tumor-bearing thymic nude mice	808, 0.5 W	[120]
Chitosan scaffolds	Graphene oxide	Human osteosarcoma cells and MC3T3-E1 pre-osteoblastic cells or human bone mesenchymal stem cells, in vivo studies for antitumor therapy, was conducted with male tumor-bearing mice	808, 2.5 and 1.2 W/cm^2^	[121]
Chitosan layer	Reduced Graphene oxide nanoflakes and IR820 dye	C26 murine colon carcinoma cells	785, 9.62 W/cm^2^	[122]
Chitosan layer	Pheophorbide	KB human oral squamous cell carcinoma cells and in vivo studies with tumor-bearing female BALB/c nude mice	680, 0.5 W/cm^2^	[123]
Chitosan coated nanosheet with tantalum oxide (TaO_2_) deposition	Molybdenum disulfide	MCF-7 human breast cancer cells	808, 0.5 W/cm^2^	[124]
Chitosan layer coating upconversion nanoparticles	Ag_2_Se	A549 human lung cancer cells and in vivo studies with tumor-bearing Kunming mice	808, 1.3 W/cm^2^	[125]
Iron crosslinked chitosan complexes	Carbon quantum dots	HeLa human cervix adenocarcinoma cells and HepG2 human hepatocellular carcinoma cells.	671, 2W/cm^2^	[126]

**Table 6 nanomaterials-10-01903-t006:** Cancer photothermal agents synthesized with nanochitosan derivatives.

Type of Chitosan Nano Polymer	Photothermal Agent	Tumor Models Used	Wavelength of Laser Source (nm)	Reference
O-Carboxymethyl chitosan-based nanoshell carrier	Photosensitizer (ABDP-SI) based on AZA-boron dipyrrolide	HeLa human cervical cancer cells	808, 1.2 W/cm^2^	[127]
Hydroxyethyl chitosan coating	Gold nanorods	MCF-7 human breast cancer cells	808, 2W/cm^2^	[128]
Dihydrophenyl/hydrazide bifunctionalized Hydroxyethyl chitosan and oxidized hyaluronic acid coating	Gold nanorods	MCF-7 human breast cancer cells	808, 2 W/cm^2^	[129]
2-hydroxypropyltrimethyl ammonium chloride chitosan coating	Copper sulfide	4T1 mammary tumor cells (mouse) and in vivo studies with Balb/c mice	808, 1.5 W/cm^2^	[130]
Thiolated chitosan coating	Gold nanorods	MDA-MB-231 human breast cancer cells	808, (1–2 W)	[131]
Carboxymethyl chitosan grafting	Reduced graphene oxide	L-929 mouse connective tissue fibroblasts	808, 1 W/cm^2^	[132]
Glycol chitosan coating	Graphene oxide-IR780 (NIR dye) and gold deposited plasmonic polylactic-co-glycolic acid nanoshells with graphene oxide	MDA-MB-231 and MCF-7 human breast cancer cells	808, 500 mW	[133]
Thiol chitosan coating	Gold nanoshells	HeLa human cervix adenocarcinoma cells and MDA-MB-231 human breast cancer cells. in vivo studies with tumor-bearing female BALB/c nude mice	808, 1.2 W/cm^2^	[134]
Glycol chitosan coating on PLGA nanoparticles	Gold nanoshells	MCF-7 human breast cancer cells.	808, 500 mW	[135]
Glycidyl methacrylate-conjugated chitosan functionalized nanogel	Gold nanorods	SCC7 mouse tumor cells, NIH/3T3 fibroblast cells and in vivo studies with male athymic nude mice	808, 4 W/cm^2^	[136]
2,3-dimethylmaleic anhydride-modified chitosan oligosaccharide-block-poly (ethylene glycol) polymer coating	Gold nanorods	MCF-7 human breast cancer cells and in vivo studies with tumor-bearing female nude mice	808, 1.5 to 2 W/cm^2^	[138]
Water soluble chitosan functionalization	Single-walled carbon nanotubes	Glioblastoma CD133^+^ and CD133^-^ cells, in vivo studies with tumor bearing BALB/c strain immunicompromised nude mice	808, 2 W/cm^2^	[139]
Thiolated chitosan surface coating	Hollow CuS nanoparticles	BALB/c mice bearing EMT6 tumor	900 nm, 2W/cm^2^	[140]
Acrylated chitosan coating	Chemically reduced graphene oxide	TRAMP-C1 mouse prostate cancer cells and Lewis lung cancer cells	808, 900 mW/cm^2^	[141]
Chitosan grafted oleic acid copolymer coating	Single-walled carbon nanotubes	HeLa human cervix adenocarcinoma cells	808, 1 W/cm^2^	[142]

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
