# Peer review of "Agricultural and Biomedical Applications of Chitosan-Based Nanomaterials"

_nanomaterials, 2020, doi:10.3390/nano10101903_

Round 1

Reviewer 1 Report

The authors present an interesting article entitled "Agricultural and Biomedical Applications of 2 Chitosan-Based Nanomaterials". The review is very clearly structured and written. I like the breadth of coverage and figures/tables that break up the text and add relevant information.

The abstract is clear and concise.
The body is clear and concise.
The conclusion is clear and concise.
The references are balanced.

Edits:

Please change "under 100 Mm salt concentration" to read "at 100 mM salt concentration"

Please ensure the charges are superscript "K+, Cl- and NO3- [67]."

Reviewer 2 Report

The review presented by the authors is a comprehensive compilation of different nanomaterials that use chitosan as a stabilizer and applications in agriculture and biomedicine. The work is well written and includes the expected aspects for a review. However, it is not a new topic and there are recent reviews that deal with these same issues, although there are none that deal with them jointly. Regarding the improvements, the work is extensive but only shows 4 figures, perhaps it would be convenient to include more figures that help to understand and illustrate the examples presented. Otherwise, I consider the work to be very good and recommend its publication.

Reviewer 3 Report

The review manuscript entitled „Agricultural and Biomedical Applications of Chitosan-Based Nanomaterials“ is a comrehensive summary of latest research in the fields of chitosan, chitosan nanoparticles, and chitosan derivatives applications in agriculture (combating the effects of high salinity, water scarcity, and heat on plant growth), food (and human) protection (making use of the antimicrobial property of chitosan to render food-borne pathogens harmless), and medical therapy (supporting photothermal treatment of cancer).

It is well written both in terms of English as well as in the structure of the manuscript. The article was pleasant to read and makes its review rather short. For the selected applications of chitosan and chitosan nanoparticles the manuscript exhibits a suitable length. I recommend to accept this paper for publication after correcting the following typos:

Table 1, reference [44]: replace multi-water carbon nanotubes by multiwall carbon nanotubes.

Lines 268, 278, 292: it should read Table 4 instead of Table 2.

Reviewer 4 Report

The review gives a good survey about the application of chitosan. In my opinion, the combination of agricultural and biomedical application does not really fit together. Maybe it would be better to separate the biomedical part from the agricultural and publish two reviews.

The references are mainly from journals without open access. Therefore, it will be difficult to access the primary literature for many readers.

Please, be carefull with formulations like: Chitosan has anticancer properties (Abstract). This is not the case. It serves as drug carrier.

The small size of the nanomaterials is advantageous in crossing the biological barriers and carrying the required molecules into various locations in animals and plants.

This can be, but in many cases it is very dangerous.

Concerning the mechanisms behind the effects of chitosan I miss more details. Example: You wrote:

According to Xiong et al., the initial stress signal may be perceived by several primary sensors leading to secondary signaling pathways that may occur at a different time and location from the primary signaling [68]. Therefore, these secondary signals may be shared between different stress response pathways providing “stress cross-protection” to the plants [68]. Please, describe the secondary signal pathways (not in details, but maybe with a figure or at least a short explanation). 
